# RESOURCE-EFFICIENT IMAGE INPAINTING

**Dharshan S. Kumar, Pranav Jeevan & Amit Sethi**
Indian Institute of Technology Bombay
Mumbai, India
{18d180009,194070025,asethi}@iitb.ac.in

## ABSTRACT

Image inpainting refers to the synthesis of missing regions in an image, which can help restore occluded or degraded areas and also serve as a precursor task for self-supervision. The current state-of-the-art models for image inpainting are computationally heavy as they are based on vision transformer backbones in adversarial or diffusion settings. This paper diverges from vision transformers by using a computationally-efficient WaveMix-based fully convolutional architecture, which uses a 2D-discrete wavelet transform (DWT) for spatial and multi-resolution token-mixing along with convolutional layers. The proposed model outperforms the current-state-of-the-art models for large mask inpainting on reconstruction quality while also using less than half the parameter count and considerably lower training and evaluation times.

## 1 INTRODUCTION

Image inpainting is a restoration process in which blemishes, holes, and other defects of an image are filled in to match the available context, thereby restoring the image's quality. A number of learning-based approaches have been proposed to tackle this problem which guarantee excellent reconstruction fidelity, but they require a large level of computational resources, especially for networks based on vision transformers (ViT) (Dosovitskiy et al., 2020). To solve this issue, a new class of networks called token-mixers, or Metaformers (Yu et al., 2021) has been proposed. WaveMix (Jeevan et al., 2023) is one such token-mixer. These alternatives consume a fraction of the resources as ViTs, but their performance is competitive with that of ViTs in simple tasks such as classification and segmentation. There is relatively less research done on their performance on image generation or restoration tasks. We investigated the application of WaveMix architectural framework to the task of image inpainting with suitable adaptations to the previously proposed architectures.

One key aspect of inpainting that differentiates it from the other tasks, such as classification and segmentation, is that it is an ill-posed self-supervised learning task, because the input and output can be generated by simply using a binary mask on any image without requiring explicit labels or annotations for supervision. It is ill-posed because any considerably large defect in any image could have multiple equally valid and contextually correct inpainted solutions. These aspects make inpainting a challenging problem, especially when the regions to be inpainted are large – the so-called *large mask* problem (Suvorov et al., 2021).

## 2 WAVEMIX INPAINTING ARCHITECTURE

The model proposed in the paper is a neural architecture that is inspired by WaveMix (Jeevan et al., 2023) as a base model. This choice is motivated by the success of WaveMix in approaching the state-of-the–art (SOTA) for different datasets on the task of parameter-efficient image classification[1]. We made the following key modifications to the original WaveMix architecture to make it applicable to the image generation task as shown in Figure 1. (1) We feed a masked image and its mask as inputs to the model, and the output is evaluated against the unmasked image. (2) The first convolutional layer was modified to reduce the image resolution by half and generate $C$ channels. (3) We added a transposed convolutional layers after the WaveMix blocks to increase the resolution of the output

---

[1] https://github.com/brohrer/parameter_efficiency_leaderboard

back to that of input. (4) Two skip connections were provided which concatenated (a) output from initial convolution layer to output from WaveMix blocks and (b) input masked image with output from final transposed convolution layer. Lastly, we replaced the loss function with a weighted linear combination of structural similarity index measure (SSIM) and mean absolute error (MAE) in the ratio 3:1 to encourage the learning of image coherence and regeneration capabilities respectively.

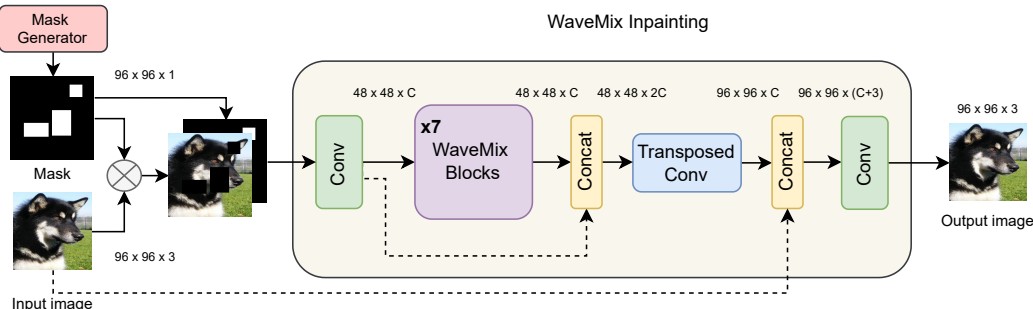

Figure 1: WaveMix inpainting architecture. WaveMix blocks are taken from (Jeevan et al., 2023). Dashed arrows represent the skip connections.

## 3 EXPERIMENTS AND RESULTS

The unlabelled images of STL-10 dataset (Coates et al., 2011a) were used for training, validation, and testing of various models. It comprises 100,000 images of $96 \times 96$ resolution. We use an 80:10:10 train, validation, test split for the experiment. The choice of this dataset over larger datasets was based on our resource-constraints. The masks for the validation and test set were pre-generated to ensure uniformity while testing and evaluating different models, but the masks for the training set were generated during training.

Table 1 illustrates the relative performance of the proposed model, a convolutional neural network (our custom architecture based on the U-Net described in Appendix A with comparable number of parameters as WaveMix) and LaMa (Suvorov et al., 2021). From Table 1, it is clear that the WaveMix model outperforms the other models on image reconstruction metrics (low Fréchet inception distance (FID) and high structural similarity index metric (SSIM)). The proposed model trains and infers faster than LaMa.

Table 1: Performance of WaveMix inpainting compared with a custom CNN and LaMa

| Model | # param ↓ | FID Score ↓ | SSIM ↑ | Train Time (s) ↓[2] | Inference Speed (s$^{-1}$ )↑[3] |
|---|---|---|---|---|---|
| Custom CNN | 12 M | 86.373 | 0.850 | **665** | **192** |
| LaMa | 27 M | 70.294 | 0.893 | 4,861 | 112 |
| WaveMix (ours) | 14 M | **64.784** | **0.902** | 1,387 | 137 |

## 4 CONCLUSION AND FUTURE WORK

This paper proposes a new architectural adaptation of WaveMix (Jeevan et al., 2022; 2023) for the less explored task of image inpainting. The performance of the proposed model is comparable to much larger models on STL-10 dataset. However, our sensitivity analysis (Appendix D) suggests that the quality of inpainting drops as the size of the mask increases. This indicates that while the model is able to capture global context, it is unable to produce semantically coherent output due to lack of generative ability. A possible solution for this problem could be to use this architecture in a generative-adverserial or diffusion setting. Thus, this paper points to the the potential of using token-mixing as alternative to vision transformers for image inpainting.

---

[2]Seconds per epoch with batch size 40

[3]Images per Second

URM Statement

The authors acknowledge that at least one key author of this work meets the URM criteria of ICLR 2023 Tiny Papers Track.

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

## A  Training Details

All training and evaluation has been performed on the STL-10 dataset Coates et al. (2011b) (under MIT Licenses). The official training dataset is split into 3 sets, namely labeled train, labeled test and unlabeled set. The images from the unlabeled set (100,000) are randomly split 80:10:10 to form the train, validation and test set for the purposes of this paper. The other 2 sets are ignored as the task in question does not require labels and they contributed a negligible number of new images compared to the unlabelled set. The FID and SSIM scores are reported on the test set of the unlabeled set. Inference throughput on a single GPU was reported in frames/sec (FPS).

*No pre-training was performed on any of the models*. The experiments in the ablation study are done on square masks of above mentioned sizes, and the final results table is generated using the randomized mask generator from Suvorov et al. (2021). The masking procedure and data augmentations are the same as used in Suvorov et al. (2021), in order to obtain reproducible and comparable results. The generated binary mask is concatenated with the masked image, bringing the total number of channels in the input to 4, which is then fed to the model. The loss function used for training the model is a weighted sum of SSIM and MAE. A batch-size of 40 was used for all experiments with STL-10 dataset. Additionally, automatic mixed precision in PyTorch was used during training.

The custom CNN used as the baseline in Table 1 is a U-net model with five down-scaling convolution layers followed by five up-scaling transposed-convolution layers. The number of parameters were chosen in such that it remain similar to the proposed model.

Due to limited computational resources (which actually inspired looking for an alternative neural framework), the *maximum* number of training epochs was set to 100. All experiments were done with a single 12 GB Nvidia 1080 GPU. For all experiments, we used AdamW optimizer ($\alpha = 0.001, \beta_1 = 0.9, \beta_2 = 0.999, \epsilon = 10^{-8}$) with a weight decay of 0.01 during initial epochs and then used SGD with learning rate of 0.001 and momentum $= 0.9$ during the final 50 epochs Keskar & Socher (2017); Jeevan & sethi (2022).

# B    ABLATION STUDIES

Multiple experiments were performed to prune the model space and get a primary estimate of the performance of various models. In the interest of time, this pruning process was done on limited data, with considerably smaller box-masks and no image augmentation. Although this may not be representative of a model trained on the entire set, since all the models are trained on the same subset, it is sufficient to notice certain patterns and the scalability of a model.

Experiments were done for 2 mask sizes, $8 \times 8$ mask and $12 \times 12$ mask. The results are as shown in Table 2 and Table 3. It is to be noted that the amount of experimentation on $8 \times 8$ is less as the models reached near-optimal results, and no further improvements were possible with available resources.

Table 2: Performance of different WaveMix architectures on inpainting with mask size of $8 \times 8$

| Model # | WaveMix blocks per module | Embedding dimension | Modules | DWT levels | Input size reduction | L1↓ | L2↓ | PSNR↑ | SSIM↑ |
|---|---|---|---|---|---|---|---|---|---|
| 1 | 3 | 96 | 1 | 1 | Not done | 0.224 | 9.455 | 41.968 | 0.988 |
| 2 | 7 | 96 | 1 | 1 | Not done | **0.134** | **5.616** | **44.154** | **0.996** |

Table 3: Performance of different WaveMix architectures on inpainting with mask size of $12 \times 12$

| Model # | WaveMix blocks per module | Embedding dimension | Modules | DWT levels | Input size reduction | L1↓ | L2↓ | PSNR↑ | SSIM↑ |
|---|---|---|---|---|---|---|---|---|---|
| 1 | 7 | 96 | 1 | 1 | Done | 0.733 | 78.542 | 26.714 | 0.825 |
| 2 | 5 | 96 | 1 | 1 | Done | 0.964 | 114.122 | 18.880 | 0.801 |
| 3 | 7 | 96 | 1 | 1 | Not done | 0.361 | 16.913 | 38.423 | 0.988 |
| 4 | 7 | 96 | 2 | 1 | Not done | **0.344** | **15.551** | **38.967** | 0.989 |
| 5 | 7 | 64 | 2 | 1 | Not done | 0.374 | 17.807 | 37.564 | **0.990** |
| 6 | 7 | 128 | 2 | 1 | Not done | 0.421 | 33.129 | 36.995 | 0.981 |
| 7 | 7 | 96 | 3 | 1 | Not done | 0.410 | 20.331 | 36.852 | 0.987 |
| 8 | 7 | 96 | 1 | 2 | Done | 0.370 | 17.672 | 38.327 | 0.988 |
| 9 | 7 | 96 | 1 | 3 | Done | 0.380 | 20.692 | 38.030 | 0.988 |

The general architecture of the model, as discussed in Section 2 is a collection of WaveMix blocks connected in series, but the number of WaveMix blocks itself is hyper-parameter. In addition to that, as mentioned in (Jeevan et al., 2023), each wave block has the ability to perform multi-levels of 2D-DWT, and the parameter "DWT levels" in Tables 2 and 3 indicates the number of levels of wavelet decomposition used. Lastly, the parameter "Input size reduction" is simple $2\times$ downsampling of images to perform experiments faster.

## C  QUALITATIVE ILLUSTRATION OF RESULTS

The images in Figure 2 show qualitative illustration of the proposed model for visual evaluation on a few sample images. One key observation from these samples is the ability of the model to incorporate context very well but its inability to produce sharp and high-quality inpainted images.

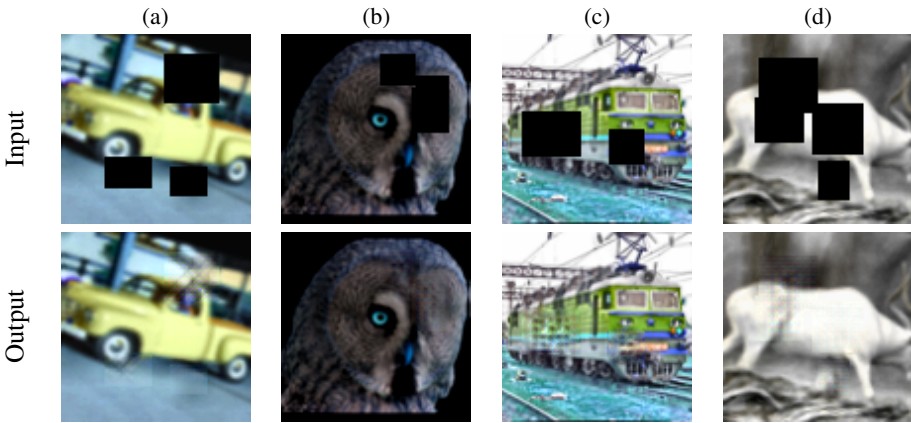

Figure 2: A visual sample of the performance of the proposed WaveMix inpainting model

## D  MODEL SENSITIVITY ANALYSIS

We studied the effect of mask size and randomization on the performance of the model. As expected, the model performed worse on bigger masks, compared to smaller masks, as in the former, the model has little to no context from the rest of the image to properly fill in the mask. Therefore, the model performance was severely affected on larger masks, but at the same time, training on too thin of a mask meant the model lost its generative abilities. This concept is illustrated in Figure 3. (The horizontal red line in $8 \times 8$ image marks one edge of the mask helps distinguish mask from background)

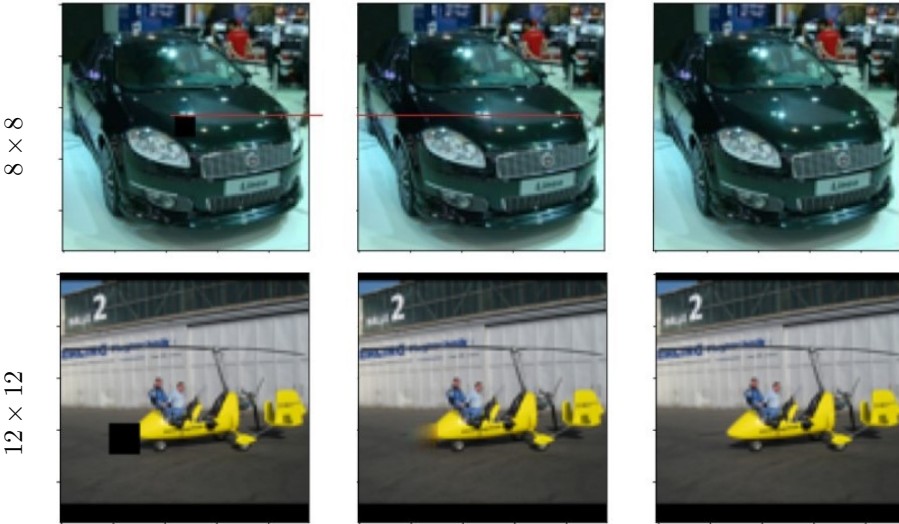

Figure 3: Illustration of model sensitivity to mask size. From left to right, masked image, inpainted model output, unmasked ground truth

Similarly, the effect of mask shape and size randomization was as expected. That is, such variation helped the model learn better and generalize to various masks, during the inference time. The effect of randomizing the masks is not seen on the training loss, but it is very much noticeable in the metrics on the test images thanks to the generalizability that it imparts.

