# OpenReview forum: "Resource-efficient image inpainting"
_ICLR.cc/2023/TinyPapers — Submitted to Tiny Papers @ ICLR 2023_

### Official Review · Reviewer_MqeL · 2023-03-30

**Confidence:** 3

**Summary Of Contributions:**

  This paper proses a resource-efficient way for inpainting by combining  U-Net architecture with Wavemix. The authors show their method outperforms traditional CNN and LaMa in terms of various metrics.

**Rating:**

Needs Clarification (NC): a submission which does not meet the reviewing criteria and needs clarification for its described problem or solution

**Strengths And Weaknesses:**

Strength
1. Resource Efficiency is a key problem and this problem is well motivated in the paper.
2. The authors provided analysis of the model with respect to mask size and randomization which provide valuable insights.

Weakness
1. This paper in its current form is not reproducible as key details are missing regarding the implementation.

**Suggested Changes:**

Suggested Changes:
1. Try your methodology on a different dataset to check model generalizability.
2. Compare with other methods such as deep image prioir and other state of the art models.
3. Connect this paper to other resource efficient architecture in the literature rather than generic comparison.
4. Provide little bit detail about token mixing algorithm and why it helps in image in-painting.

---

> ### Author Response · Authors · 2023-06-01
> **Response**
>
> 1. The review is appreciated, but due to resource and time constraints the study could only be done on the smaller STL-10 dataset, and all results and ablation are reported in the result and appendix section.
> 2. The lama model was chosen in order to stick to the theme of the paper pertaining to parameter efficiency and low inference time costs.
> 3. Since the approach is relatively novel it holds only vague similarities to lama and others in the architectural design, therefore the lack of connection.
> 4. The token mixing algorithm used in the paper, called WaveMix is a meta-former module, in which the mixing of information happens using a multi-level two-dimensional discrete wavelet transform and through experimentation it was found to consume significantly lower computational costs and memory footprint for multiple vision tasks. This paper is an extension of that algorithm for image generation task of inpainting.

---

### Official Review · Reviewer_cxPf · 2023-03-31

**Confidence:** 4

**Summary Of Contributions:**

This paper proposes a Discrete-Wavelet-Transform (DWT) based token mixing block for unsupervised ill-posed image inpainting. Empirical results show that the proposed model outperforms LaMa in training time and performance.

**Rating:**

Great Start (GS): a submission which meets some of the reviewing criteria but has room for improvement

**Strengths And Weaknesses:**

### Strengths

1. Efficient token mixing block. The proposed DWT-based token mixing block is parameter-efficient and fast to train.

2. Clear explanation of methods. The paper presents a clear explanation of the proposed method, making it easy to understand.

3. Visualization of inpainting images. The proposed model clearly recovers the missing parts.

### Weaknesses

1. Writing Clarity. The paper could be carefully polished.

2. Lack of ablation study. While the paper introduces several modifications over the baseline, including the DWT block, transpose convolutional layers, and combined loss, the authors do not conduct an ablation study to demonstrate their utility.

**Suggested Changes:**

1. Polishing the writing. The authors are encouraged to carefully polish the paper to improve its clarity. Authors could use more text to describe their methods and motivation. The authors should provide more details about the model configuration and training settings in the appendix for better reproduction. There are typos in the abstract, like ``Discreet'' (Discrete).

2. Conducting an ablation study. The authors should conduct an ablation study to demonstrate the utility of the proposed modifications over the baseline.

---

> ### Author Response · Authors · 2023-06-01
> **Response**
>
> 1. The paper has been re-written in a more legible and detailed manner to improve the clarity.
> 2. Ablation study and training details have been included in the appendix for reproducibility

---

### Meta-Review · Area_Chair_3azQ · 2023-04-08

**Recommendation:** Invite to revise
**Confidence:** 4

**Metareview:**

Based on the two reviews, the proposed paper presents a Discrete-Wavelet-Transform (DWT) based token mixing block for unsupervised ill-posed image inpainting. The paper highlights that the proposed model outperforms LaMa in training time and performance, by combining the U-Net architecture with Wavemix, which addresses the resource efficiency problem in inpainting.

Strengths of the paper include the proposed DWT-based token mixing block being efficient and fast to train, clear explanation of methods, and the visualization of inpainting images. The authors also provide valuable insights into the model's analysis with respect to mask size and randomization.

Weaknesses of the paper include the writing clarity that could be improved, and missing key details regarding the implementation which make it difficult to reproduce the results. Additionally, the paper lacks an ablation study to demonstrate the utility of the various modifications introduced over the baseline.

Overall, the paper provides a promising approach to addressing the resource efficiency problem in inpainting, and the proposed DWT-based token mixing block could potentially be of interest to researchers working on similar problems.


**Summary:**

This paper proposes a Discrete-Wavelet-Transform (DWT) based token mixing block for unsupervised ill-posed image inpainting.

**Reason For Not Giving A Higher Recommendation:**

 It can be further improved when incorporating the suggested changes by the reviewers.


**Reason For Not Giving A Lower Recommendation:**

N/A

---

### Decision · Program_Chairs · 2023-04-09

Revision accepted; invite to archive

---

> ### Comment · Area_Chair_3azQ · 2023-06-06
> **Meet threshold for archival**
>
> This work meets the threshold for archival, contents the URM statement and is deanonymized.